# Metabolomics Insights into Inflammatory Bowel Disease: A Comprehensive Review

**DOI:** 10.3390/ph14111190

**Published:** 2021-11-20

**Authors:** Laila Aldars-García, Javier P. Gisbert, María Chaparro

**Affiliations:** Gastroenterology Department, Hospital Universitario de La Princesa, Instituto de Investigación Sanitaria Princesa (IIS-IP), Universidad Autónoma de Madrid (UAM), Centro de Investigación Biomédica en Red de Enfermedades Hepáticas y Digestivas (CIBEREHD), 28006 Madrid, Spain; laila.alga@gmail.com (L.A.-G.); javier.p.gisbert@gmail.com (J.P.G.)

**Keywords:** inflammatory bowel disease, metabolomics, microbiota

## Abstract

Inflammatory bowel disease (IBD) is a chronic, complex relapsing disorder characterised by immune dysregulation, gut microbiota alteration, and disturbed intestinal permeability. The diagnosis and the management of IBD are challenging due to the recurrent nature and complex evolution of the disease. Furthermore, the molecular mechanism underlying the aetiology and pathogenesis of IBD is still poorly understood. There is an unmet need for novel, reliable, and noninvasive tools for diagnosing and monitoring IBD. In addition, metabolomic profiles may provide a priori determination of optimal therapeutics and reveal novel targets for therapies. This review tries to gather scientific evidence to summarise the emerging contribution of metabolomics to elucidate the mechanisms underlying IBD and changes associated with disease phenotype and therapies, as well as to identify biomarkers with metabolic imbalance in those patients. Metabolite changes during health and disease could provide insights into the disease pathogenesis and the discovery of novel indicators for the diagnosis and prognosis assessment of IBD. Metabolomic studies in IBD have shown changes in tricarboxylic acid cycle intermediates, amino-acid and fatty-acid metabolism, and oxidative pathways. Metabolomics has made progress towards identifying metabolic alterations that may provide clinically useful biomarkers and a deeper understanding of the disease. However, at present, there is insufficient evidence evaluating the predictive accuracy of these molecular signatures and their diagnostic ability, which is necessary before metabolomic data can be translated into clinical practice.

## 1. Introduction

Inflammatory bowel disease (IBD), which includes Crohn’s disease (CD) and ulcerative colitis (UC), is a chronic disorder of unknown aetiology involving a pathological response of both the innate and the adaptive immune systems, resulting in chronic inflammation of the digestive tract. IBD mainly affects young patients and is considered a disabling disease [1]. Currently, there is no curative treatment for IBD; therefore, the therapeutic goal is to control the inflammatory process to prevent the onset of symptoms and the development of complications.

The complexity and costs associated with the treatment of IBD, in addition to its great social burden, make this disease important for health systems. Specifically, in the USA, it is one of the five pathologies with the greatest social burden, with a mean annual cost (1998–2000) of 1.7 billion USD in healthcare services [2]. In Europe, the annual costs (direct and indirect) associated with IBD exceed 25,000 million EUR [3].

The main difficulty in the treatment of IBD is that its aetiology is not exactly known. This is because the pathogenesis of IBD is highly complex, being the result of the interaction among environmental factors, the gut microbiota, the immune system, and the genome; ultimately, an aberrant immune response to commensal microbiota is triggered in some individuals [4]. Each of these factors (exposure to environmental factors or exposome, the gut microbiota or microbiome, the immune system or immunome, and the genetic load or genome) is in itself a very complex entity. Although none of them can individually cause the disease, their interaction known as the “IBD integrome” is crucial for disease pathogenesis [4].

For instance, the human genome carries crucial genetic information to trigger this disease. However, knowledge of this information alone is insufficient to elucidate the physiological processes involved. Other comprehensive tools in molecular biology have emerged with the aim of building on our knowledge of the genome to understand transcription and the resulting protein activity, as well aa elucidate the absolute extent of physiological pathways. These tools are collectively termed “functional genomics”, “systems biology”, or more colloquially “omics”. The development of new “omics” technologies would allow a new approach to this biological complexity, to identify and understand the cellular and molecular pathways involved in the development of the disease. Different “omics” methodologies are currently increasingly recognised as a powerful tool for an increased genome-to-metabolite characterisation of biological processes in gastrointestinal diseases [5,6,7]. Genomic and protein data mainly indicate the potential for specific metabolic functions, whereas metabolomics integrates the effects of gene regulation, post-transcriptional regulation, and pathway interactions; in other words, the read-out of metabolomics conveys more information about the phenotype [5].

The metabolome can be described as the study of the complete expression and biological function of low-molecular-weight molecules (less than 25 kilodaltons) within a biological system [8]. Metabolomic analysis refers to the comprehensive study of the small molecules present in biological samples, using technologies which enable the analysis of multiple metabolites with high sample throughput. Therefore, proper analytical techniques are required for the rapid and simultaneous determination of a broad range of these small molecules. Hence, spectroscopic and spectrometric techniques are used, such as nuclear magnetic resonance (NMR) spectroscopy or mass spectrometry (MS) [9].

Metabolites are generated within the organism by the activity of enzymes coded by genes included in its genome. Therefore, genetic mutations affecting the enzymatic function of certain proteins will alter the metabolite profile of the organism [9]. These metabolite changes may be detected by metabolomic analysis and may provide insights into disease pathogenesis [10].

The aim of the present review is to summarise the emerging contribution of metabolomics to elucidate the mechanisms underlying IBD and to identify biomarkers with metabolic imbalance in those patients.

## 2. Methods

An electronic search was conducted using the MEDLINE database via PubMed to identify published articles on the IBD metabolome up to June 2021.

Included search terms were related to (1) metabolomics, (2) IBD, (3) inflammation, and/or (4) biomarkers. Studies were eligible for inclusion if they assessed the metabolome in IBD patients and NMR-based or MS-based metabolomic approach.

Additional articles were identified by reviewing the references of examined publications.

Paediatric studies, animal studies, in vitro model studies, review articles, and abstracts without full texts were not included.

## 3. Results and Discussion

### 3.1. Biosamples in Metabolomic Analysis

Metabolomic studies can be conducted in a variety of biological fluids and tissues, from easily accessible biofluids such as blood, urine, saliva, or faeces to more invasive samples such as organs, tissues, or even cells. Metabolomics results clearly depend on the biologic matrix chosen, containing information related to inherent parameters, such as the genotype, and to environmental factors, including the diet, xenobiotics, and the gut microbiota [11].

In this respect, serum and plasma show a largely different metabolite profile from faecal and urine samples, because serum or plasma profiles reflect changes in the host’s metabolism rather than those in the gut microbial activity, diet, or xenobiotics [5]. The microbial metabolism is more likely reflected in the faecal and, to lesser extent, in the urinary metabolome. In addition, metabolites derived from the host metabolism are returned to the gut via biliary excretion and are then metabolised by the microbiota, representing host–microbial cometabolites [12]. Many microbial metabolites are absorbed from the colonic lumen and excreted in urine, either as such or after being processed by human enzymes. Urinary and faecal profiles are more variable due to the susceptibility of these biofluids to environmental conditions. Urinary profiles contain human and human–microbial cometabolites, whereas serum profiles seem to be less influenced by bacterial metabolism and present lower inter-subject variability [13]. Table 1 summarises the main findings in the IBD metabolome as compared to controls.

Several studies, irrespective of the biological matrix analysed, have shown the potential of metabolomics to differentiate between healthy subjects and patients with IBD, as well as between IBD subtypes, and several metabolites have been identified with potential use as biomarkers (Appendix A). In general, these studies have a number of limitations, such as the small sample size or the lack of a valid gold standard (such as endoscopic evaluation) to assess disease activity, and most of them did not take into account previous and current treatments (which can modify the biological state), in addition to a lack of information about the accuracy of the biomarkers, a lack of longitudinal studies, and heterogeneous results that have not been validated in independent cohorts.

### 3.2. Methodology to Study the Metabolome of Biological Samples

To date, the two main technical approaches for unbiased study of the metabolome currently used are NMR and MS. These techniques allow for the analysis of multiple metabolites in biological samples and have become invaluable tools for metabolomic analysis [60].

MS, in essence, measures the molecular mass of chemical compounds and their fragmentation products. This technique is based on the ionisation and fragmentation of different compounds into smaller molecules that can be identified and quantified. To enable the separation of metabolites prior to MS analysis, gas chromatography (GC) and liquid chromatography (LC) are commonly used interfaces, enhancing the detection of individual molecules [60].

NMR measures the magnetic resonance of atomic nuclei in molecules. The resonance frequencies of nuclei are influenced by the nature and number of surrounding nuclei; the frequency and the pattern of resonance provide a spectroscopic signature for each metabolite as a function of its chemical structure. Of particular interest in clinical metabolomics are ^1^H, ^13^C, ^19^F, and ^31^P [61,62]. The most utilised NMR-active isotope in these studies is the proton (^1^H), which enables the detection of all proton-containing low-molecular-weight metabolites with a limit of detection of approximately 10 μM. Furthermore, ^13^C-NMR, following incorporation of a labelled precursor, enables the interrogation of cellular pathways such as glycolysis and the tricarboxylic acid (TCA) cycle. ^31^P-NMR allows for the detection of metabolites that provide insight into energy or phospholipid metabolism [61,63]. Unlike MS, identification of metabolites by NMR can be performed without prior separation of compounds in the sample.

To date, NMR has been the dominant platform for metabolomic analysis. The main advantage of NMR is that it is highly quantitative and reproducible along with minimal sample preparation requirements, in addition to the non-destructive nature of the platform, which allows for further analysis of the samples. The most significant drawbacks of the NMR platform are the lower level of sensitivity compared to MS and higher costs. On the other hand, MS presents higher sensitivity but requires extra steps for sample preparation, such as separation or derivatisation, compared to NMR [60].

Once the biological samples have been analysed, data pre-processing involves extracting the relevant chemical signals and normalising for factors that could affect the statistical analysis, i.e., analytical processing variability or sample characteristics [64]. Then, univariate and multivariate statistical techniques are employed to determine which metabolites differ between sample groups [65].

Examples of commonly used multivariate discriminant tests in metabolomic studies are partial least-squares discriminant analysis (PLS-DA) [24,33,34,41,46,52,53] and principal component analysis (PCA) [13,16,31,34,46,52]. PLS-DA, a multivariate dimensionality-reduction tool, is a supervised classification method, in which the samples are designated into their classes for comparison. In contrast to PLS-DA, PCA is thought to be an unsupervised classification method, as it considers the variation in the data without a priori designation of samples into their classes. Thus, PCA and untargeted methods in general are useful for identifying unexpected class groups or trends without any preformed ideas.

### 3.3. Metabolomics Use to Distinguish IBD (CD and UC) Patients from Healthy Controls

Metabolomic profiles have been used to discriminate IBD patients from controls using different biological matrices, i.e., serum, plasma, urine, stool, or intestinal tissue. Clear differences have been observed for all biological matrices in the metabolomic profiles between individuals with IBD and healthy controls (HC) (Table 1). The main changes of the metabolomic profile found in the literature involve gut microbiota derivates, alterations due to the impairment in the gut barrier, and energy metabolism perturbations; therefore, in this section, we briefly describe such changes.

#### 3.3.1. Gut Microbiota Metabolites

The gut microbiota is involved in IBD pathogenesis, and it has been hypothesised that gut dysbiosis could either be a cause or a consequence of the disease [66], yet to be determined. The relationship of gut microbiota alterations with the individual’s metabolic state was evidenced by Yap et al., who showed how perturbations of the gut microbiota by antibiotic administration to mice affected the host’s systemic metabolic phenotype [67]. Microbes transform complex carbohydrates from the diet, producing formate, lactate, pyruvate, or succinate and short-chain fatty acids (SCFAs; mainly butyrate, acetate, and propionate). The microbiota also degrades proteins, producing branched-chain fatty acids (BCFA), amino acids, amines, harmful phenolic compounds, ammonia, and hydrogen disulphide [68,69]. The gut microbiota can also modify host-derived metabolites, such as bile acids (BAs), intermediates of the TCA cycle, and cholesterol metabolites, and it can synthesise de novo metabolites, such as adenosine triphosphate [70].

Some of the aforementioned metabolites have been proven to be altered in IBD patients compared to controls, reinforcing the gut dysbiosis hypothesis in these patients. One of the main traits of IBD gut dysbiosis is the decrease in *Clostridium* cluster IV and *Faecalibacterium prausnitzii* [71]. These groups of bacteria exert many beneficial effects on our intestinal homeostasis [72] and have been positively correlated with urinary hippurate [73]. Urinary hippurate is a product of the microbial metabolism of certain dietary compounds to benzoic acid, with subsequent renal and hepatic conjugation of benzoic acid with glycine [46]. Hippurate levels have been found to be significantly lower in CD and UC patients compared to controls [39,41,46,47,48]. Similarly, *p*-cresol sulphate (product of the bacterial metabolism of tyrosine) is produced by certain bacteria, mainly by *Clostridia* spp. [74]. Williams et al. reported significantly reduced levels of *p*-cresol in the urine of CD patients as compared with healthy subjects [46]. Furthermore, tyrosine was found increased in faecal samples of CD patients as compared with healthy subjects [52,55,59], suggesting that the microbiota involved in its degradation is impaired. Indolic derivatives, produced from tryptophan by commensal bacteria, have also been reported to be altered, including indole-3-propionic acid (decreased), indole-3-acrylic acid (decreased), and 3-methylindole (increased) [29]. Studies have shown how indolic derivatives are implicated in intestinal inflammation, exerting a critical role in the regulation of immunity and inflammation [75,76].

In addition, some researchers analysed the relationship between the metabolome and the microbiota through metabolomic and microbiota analyses. Le Gall et al. correlated the gut microbiota profile and metabolite composition in UC patients. They used canonical correlation analysis of NMR and PCR-denaturing gradient gel electrophoresis data to separate UC patients from controls, finding strong evidence for a direct causal link between the microbial composition in both healthy and diseased conditions and the corresponding faecal metabolite profiles [54]. Santoru et al. observed significantly different faecal metabolic and microbial profiles between IBD patients and control subjects. They found higher abundance of certain pathogenic bacterial genera in IBD, which may have caused significantly altered levels of host–microbial cometabolites, including biogenic amines, amino acids, lipids (significantly increased in IBD), and two B group vitamins (significantly decreased in IBD) [56]. Likewise, the composition of the gut microbiota was associated with serum levels of tryptophan (significantly lower in IBD patients) [44] and urinary levels of levoglucosan (increased in postoperative patients who recurred and correlated with Bacteroidales and Gammaproteobacteria) [20]. Supporting these observations, Franzosa et al. performed untargeted LC–MS metabolomic and shotgun metagenomic profiling of cross-sectional stool samples from IBD patients and control subjects. They identified 122 robust associations between differentially abundant microbes and metabolites, indicating possible mechanistic relationships that are perturbed in IBD [58].

To date, several abnormalities in fatty acid (FA) metabolism have been identified among IBD patients. FAs play important roles in the regulation of physiologic and metabolic pathways. They have an important role in inflammation; some of them exhibit proinflammatory functions, while others exhibit anti-inflammatory functions. Impairments in their normal levels modify the inflammatory response [29]. Specifically, SCFAs play an important role in maintaining intestinal homeostasis. Butyrate is an important energy source for intestinal epithelial cells, strengthening gut barrier function and exerting important immunomodulatory functions including the induction of regulatory T (Treg) cells and mucus production to downregulate inflammatory signalling pathways. The study of IBD patients’ faeces and urine has consistently revealed reduced levels of SCFAs compared to the control population [47,52,55,57,58,59]. These changes are more marked in CD than in UC patients [52], pointing to higher inflammation and dysbiosis in CD. Notably, Lloyd-Price at al. performed faecal metabolomic analysis in IBD patients and found that SCFAs were generally reduced in dysbiotic patients, providing further evidence that dysbiosis assessment is specifically relevant in IBD [57].

#### 3.3.2. Metabolic Alterations due to Compromised Intestinal Barrier

The gut epithelial barrier is compromised by inflammation, leading to the malabsorption of nutrients, including amino acids. Amino-acid malabsorption results in higher levels of amino acids in faeces and lower levels in the urine and blood of IBD patients. Several studies support this hypothesis, reporting increased levels of amino acids in stool samples of IBD patients compared to controls [39,52,53,54,55], together with decreased levels of amino acids in the serum of IBD patients [25,27,29,33,39,44]. Such results suggest that patients suffering IBD exhibit an impaired absorption of amino acids in the gut, likely due to inflammation and mucosal damage [24].

For instance, employing ^1^H-NMR spectroscopy and multivariate pattern recognition techniques, the faecal metabolomic profile in IBD patients revealed alterations in several metabolic pathways of amino acids such as lysine, and branched-chain amino acids (isoleucine, leucine, and valine) [52]. In another study, Scoville et al. reported a decrease in essential (leucine, lysine, and valine), semi-essential (arginine and glutamine), and nonessential (serine) amino acids in CD serum samples compared to control subjects, but no differences in serum amino acids in UC compared with controls [25]. Consistently with previous studies, Notararigo et al. found lower levels of creatine, proline, and tryptophan in UC patients, reflecting a deficit in the absorption of essential amino acids in the gut [27].

This disturbance in amino-acid metabolism could have relevant consequences in IBD prognosis, due to the pivotal role of amino acids as regulators of the inflammatory process; in this sense, the amino-acid profile is expected to modulate inflammation in the gut [77].

#### 3.3.3. Energy Metabolism Alteration

In IBD, perturbations have been identified in energy metabolism, including carbohydrate, amino-acid, lipid, and vitamin metabolism. Metabolic signatures of energy metabolism alteration were recently observed in a study by Scoville et al., where a number of lipid-, and TCA-related metabolites were significantly altered in IBD patients’ serum samples [25]. Other studies also support these observations of altered energy metabolism including metabolites involved in the amino-acid cycle [15,33,39,41,47,49,52,53,55,75] and TCA cycle [15,18,39,41,47,48].

Consistently, all studies report a significant decrease in TCA intermediates, including citrate, aconitate, α-ketoglutarate, succinate, fumarate, and malate in IBD compared to controls, in serum, plasma, urine and tissue, whereas no change was observed in faecal samples. TCA cycle metabolites have diverse nonmetabolic signalling roles with important effects on physiological state. Some TCA metabolites can alter the response of both the innate and the adaptive immune systems, including acetyl-CoA, succinate, α-ketoglutarate, or fumarate. In addition, succinate and fumarate promote tumourigenesis [78]. Therefore, the identified alterations may be implicated in the impaired immune response in IBD patients. Nevertheless, further studies are required to investigate such association.

Impairment in energy metabolism is also reflected in serum ketone bodies, which have been found consistently increased in IBD [40,41,45,47]. Ketone bodies are mainly produced in the liver from fatty-acid oxidation derived acetyl-CoA. In this context, glucose was reported to be increased in the serum of IBD patients [33,40,41], reflecting the inability of the body to use glucose, which may lead to an increase in ketone bodies.

### 3.4. Metabolomics Use to Distinguish CD from UC, and the Different IBD Subclassifications

After IBD is diagnosed, differentiating UC from CD is challenging and may produce ambiguous results. Accordingly, a correct diagnosis is of great importance to provide adequate treatment for patients. However, there is no single standard method for differential diagnosis of CD/UC, and some patients might be misdiagnosed. To address some of the current controversies in IBD diagnosis, metabolomic profiles could shed light on the challenges in discriminating these two IBD types. Several metabolomic studies have tried to identify differences in the metabolomic profile between CD and UC patients [14,18,25,33,38,39,46,52].

One of the first metabolomic approaches to characterise the metabolome of IBD colonic biopsies was performed by Bezabeh et al. [14]. The authors constructed a classifier model based on the metabolomic pattern by using linear discriminant analysis, achieving a classification accuracy of 98.6% in distinguishing between UC and CD. Williams et al. showed that metabolites related to the gut microbiota were significantly altered in CD in comparison with UC and controls [46]. Similarly, gut microbiota faecal metabolites showed higher disturbance in CD in comparison with UC, indicating that the inflammatory burden is higher in CD than in UC [52]. Another study based on ^1^H-NMR characterised IBD patients’ serum to distinguish CD and UC cohorts from each other. They used discriminant analysis, obtaining significant predictive accuracy, highlighting differences in lipid and choline metabolism between these two groups [33]. Similarly, in serum, Scoville et al. found that lipid metabolism and amino-acid metabolism were the main contributors to distinguish between CD and UC [25]. Ooi et al. demonstrated that serum amino acids and TCA cycle-related molecules showed different profiles in CD and UC, as compared to control subjects [18]. Metabolic profiling of serum and plasma by ^1^H-NMR spectroscopy and multivariate analysis discriminated between UC and CD patients; however, lower predictability was obtained compared to discrimination between CD or UC and control subjects [18,33,39].

Additionally, metabolomic studies aiming to discriminate between the different IBD subclassifications, i.e., CD phenotypes, location, or severity and UC location or severity, are scarce. This aspect is of great interest to understand the differences in mechanisms underlying different IBD phenotypes, which might lead to propose new therapeutic targets and drug selection based on patients’ characteristics. In this respect, Jansson et al., analysing the faecal metabolome, were able to distinguish between predominantly ileal CD and predominantly colonic CD patients. They found increased amounts of fatty acids such as arachidonic acid, linoleic acid, oleic acid, stearic acid, palmitic acid, and 6*Z*-, 9*Z*-, and 12*Z*-octadecatrienoic acid in ileal CD versus colonic CD and control subjects. Interestingly, they developed a PLS model that revealed a clear separation between the individuals with ileal CD versus colonic CD and healthy individuals, and between colonic CD and healthy individuals [53]. Recently, Notararigo et al. used NMR to detect unique biomarkers for different IBD classes, reporting higher levels of homoserine–methionine and isobutyrate in ileocolonic CD. In addition, among the three CD location types, according to the Montreal classification (ileocolic, ileal, and colonic), only ileocolonic showed significant differences versus control subjects [27].

### 3.5. Metabolomic Differences Based on Disease Activity and Predictors of Relapse

During the last few decades, several studies reported differences in IBD metabolome based on disease activity [15,16,19,23,25,26,28,29,30,34,41,42,43,45,48]. However, most of these studies had a cross-sectional design comparing the metabolome from patients with active disease and quiescent disease, which may only reflect individual differences in the metabolome rather than disease activity. Hence, longitudinal studies would provide more valuable information related to the pathophysiological understanding of the disease and flare up prediction.

Using ^1^H-NMR spectroscopy, Balasubramanian et al. studied the metabolism of the colonic mucosa of CD and UC patients with active and quiescent disease, as well as control subjects. During active phase, significantly lower concentrations of amino acids (isoleucine, leucine, valine, alanine, glutamate, and glutamine), membrane components (choline, glycerophosphorylcholine (GPC), and myo-inositol), lactate, and succinate were observed compared to control subjects, whereas, in remission, their concentrations were similar. In addition, formate levels, in colonic tissue, were found to be significantly lower in patients with active UC compared with patients with CD in an active state, but not in remission [15]. Consistently with Balasubramanian et al., Bjerrum et al. performed metabolomic profiling in mucosal colonic biopsies, colonocytes, lymphocytes, and urine from patients with UC and control subjects, using NMR spectroscopy and multivariate analysis. Significant differences between control subjects and active UC were found in the metabolomic profiles of biopsies and colonocytes. Biopsies from patients with active UC showed higher levels of antioxidants and of a range of amino acids; on the other hand, they found lower levels of lipid, GPC, myo-inositol, and choline. Only 20% of quiescent patients had similar profiles to those in an active state. However, they were not able to differentiate between active UC, quiescent UC, and control subjects using urine samples. On the contrary, higher levels of amino acids were detected in colonic tissue in the same conditions [16].

In addition, patients with active disease, both UC and CD, showed an increased level of α-glucose when compared to control subjects in the colonic mucosa of active UC patients compared to the control subjects, indicating the inability of the colonic mucosal cells to utilise glucose as energy source, which may lead to loss of mucosal integrity [16].

Alonso et al. showed that urinary citrate concentration was significantly lower in CD patients with more severe inflammation, pointing out this metabolite as a possible biomarker of disease activity, specifically in CD patients [48]. Accordingly, Dawiskiba and colleagues conducted a metabolomic analysis in serum and urine samples of both active and quiescent IBD patients and control subjects. Comparing active vs. quiescent, in serum samples, they reported a reduced level of low-density lipoproteins and increased levels of *N*-acetylated compounds and phenylalanine. In urine, they reported lower levels of acetoacetate and increased levels of glycine [41].

Notably, using plasma metabolites analysed by NMR and unsupervised analysis, Probert et al. were able to separate metabolite profiles on the basis of UC endoscopic severity. Metabolite levels involved in this discrimination included decreases in lipoproteins and increases in isoleucine, valine, glucose, and myo-inositol in patients with high compared to low endoscopic index [26]. High correlation between disease activity and levels of tryptophan and its metabolites was found in serum samples [28,44], which could be related to an altered gut metabolism and impaired absorption of tryptophan by epithelial cells.

Reinforcing previous results, Bjerrum et al. and Lai et al. also hypothesised that metabolomics is not only able to distinguish between IBD patients and control subjects, but also to provide distinct metabolic patterns depending on active and quiescent states of the disease. Metabolic profiles were significantly different between active/quiescent UC/CD and control subjects, including metabolites of the energy metabolism (e.g., β-oxidation of fatty acids and pyruvate metabolism), lipid signalling cascades (e.g., DHA), and amino acids (e.g., l-tryptophan), suggestive of an intense inflammation-driven energy demand in active disease [23,29].

However, Scoville et al. [25] and Wilson et al. [43] did not find differences in the metabolomic profile of serum samples in clinically active CD compared with quiescent CD. Furthermore, few differences between active and quiescent disease were found by Sun et al., who only found one significantly different metabolite between the active and quiescent states, whereas 34 metabolites were found to be different between active UC and control subjects and 38 between quiescent UC and control subjects [30].

Studies that prospectively monitor IBD metabolome are scarce; nevertheless, some evidence was reported on the ability to predict relapse [42,45]. Attempting to assess clinical relapse, Keshteli et al. analysed urinary and serum metabolites in a 12 month follow-up study of UC patients in remission. In comparison to UC patients who were still in remission during follow-up, patients with clinical relapse had significantly higher levels of *trans*-aconitate (urine), 3-hydroxybutyrate (serum), acetoacetate (serum), and acetone (serum), and lower levels of acetamide (urine) and cystine (urine) [45]. Cystine is involved in glutathione biosynthesis, which may reflect the oxidative damage associated with relapse, while acetamide, obtained from high-fibre diets, has antimicrobial, anti-inflammatory, and antibiotic functions. Similarly, the prediction of relapse was investigated in UC patients in remission prospectively assessed for plasma amino-acid concentrations for up to 1 year. The researchers concluded that plasma amino-acid profiles in UC patients in clinical remission can predict the risk of relapse within 1 year, as histidine levels were associated with increased risk of relapse [28].

### 3.6. Changes in the Metabolome in Response to Treatment

Changes in the metabolome associated with response to treatments have been less studied. In this sense, longitudinal studies are key for correlating a metabolomic profile with the response to treatment and ongoing disease activity. Few longitudinal studies (excluding interventional studies) have been performed in IBD patients and control subjects, revealing that adequate monitoring is crucial for identifying disease relapse and response to treatments [23,27,36].

Metabolomic profiling has the potential to predict response to therapeutic agents. Significant differences were observed in the concentration levels of some essential amino acids between control subjects and IBD patients undergoing thiopurine treatment [49]. Regarding anti-TNF therapy, IBD patients not responding to infliximab were identified as a potentially distinct group according to their metabolic profile, although no applicable response biomarkers could be identified in serum [23]. Interestingly, in a recent study, ileocolic CD patients in remission due to anti-TNF treatment showed an increase in serum SCFA with respect to control subjects [27], pointing to potential serum response biomarkers. Another study conducted in serum, urine, and faeces obtained promising results for the prediction of response to anti-TNF therapy in CD. Histidine and cysteine were identified as response biomarkers, in serum and urine, and the receiver operating characteristics curve for treatment response developed using serum BAs demonstrated a 0.74 ± 0.15 predictive ability for anti-TNF response in CD [36]. Employing a transversal approach, a recent study showed how CD patients receiving anti-TNF had an increase in total serum BAs compared to unexposed patients. Similarly, Roda et al. found that anti-TNF treatment modified the secondary BA serum profile in IBD patients [31]. In CD, primary BAs are increased in patients with prior ileocolonic resection, while a nonsignificant trend towards lower secondary BAs is observed in surgery samples. In UC, there is a similar trend towards increased primary BAs in patients with colectomy and J pouch; however, no significant changes are found in secondary Bas [51]. All these results point to BAs as potential indicators of treatment response and disease progression.

The urinary metabolome of CD patients who undergo ileocolonic resection revealed that CD patients with endoscopic disease recurrence after surgery have a unique urinary metabolomic fingerprint, i.e., increased levoglucosan concentration, that can differentiate them from CD patients who are in endoscopic remission after ileocolonic resection [20]. Reinforcing these results, Fang et al. recently evidenced that several metabolites were differentially abundant in individuals with prior surgery; most of these were BAs, in addition to tyrosine and glutamic acid, which were less abundant in subjects with surgery [51].

Treatment responses need to be closely monitored to avoid complications and to determine treatment effectiveness. Recent studies have further demonstrated the relationship between IBD metabolome dynamics and response to treatment, but this area still needs to be explored.

## 4. Final Remarks and Future Perspectives

IBD is a heterogeneous and multifactorial disease affected by various pathophysiological pathways; therefore, a single biomarker approach cannot be postulated as ideal for clinical application. Metabolomics allows the measurement of hundreds of metabolites in biological samples and, thus, characterisation of each patient by their own metabolic profile. These profiles could improve our understanding of this complex disease and potentially also improve the diagnosis and management of IBD, thereby facilitating the implementation of metabolomic personalised medicine.

Recent technological advances have boosted the capacity to study the metabolite profile of biological matrices associated with IBD. Metabolomic profiling of faeces and urine has revealed imbalances in gut microbiota, emphasising the critical role of dysbiosis in IBD. Furthermore, the gut barrier is compromised in these patients, leading to impaired nutrient absorption, as well as the maintenance of a chronic immune response in the intestinal environment. Additionally, changes in TCA cycle intermediates, amino-acid and fatty-acid metabolism, and oxidative pathways lead to a loss of energy homeostasis that may play a significant role in the pathogenesis of IBD. All these changes in IBD patients make it possible to identify potential markers within the different disease types and states, as well as optimal therapies. Among the biological matrices, serum is easily accessible and offers good sensitivity, a broad dynamic range of metabolites based on disease state, and lower inter and intrasubject variability (i.e., it is less affected by environmental conditions than faeces and urine), representing a successful candidate in metabolomic studies.

Metabolomics strongly complements the previously established genomic, transcriptomic, and proteomic technologies. These different levels of omics data from patients with IBD give us the opportunity to investigate the crosstalk among the key players in IBD pathogenesis with high translational potential. In addition, longitudinal studies are key to understand the global evolution of biological processes; therefore, longitudinal multiomic data are essential to advance in the knowledge of IBD pathophysiology, providing a more comprehensive view of biological processes, as well as to diagnose, predict the response to therapeutic agents, and monitor the disease. However, at present, limited knowledge is available regarding the predictive value and diagnostic ability of metabolic biomarkers using either a single metabolite or a set of metabolites.

Access to large-scale omics datasets will lead to the emergence of systems biology approaches to advance our understanding of IBD processes, thereby uplifting the field of predictive, preventive, and personalised medicine. However, this promising perspective holds several challenges regarding the development of adequate computational and informatics frameworks. There is a need to address the integration of omics datasets for the development of standardised analytical workflows that can be implemented by the omics research community.

## Figures and Tables

**Table 1 pharmaceuticals-14-01190-t001:** Main findings in the IBD metabolome as compared to controls.

Category			No of Studies	References
Biosamples in Metabolomic Analysis		Tissue	9	[14,15,16,17,18,19,20,21,22]
		Blood	25	[18,22,23,24,25,26,27,28,29,30,31,32,33,34,35,36,37,38,39,40,41,42,43,44,45]
		Urine	10	[16,36,39,41,45,46,47,48,49,50]
		Faeces	11	[30,36,51,52,53,54,55,56,57,58,59]
Methodology		NMR	21	[14,15,16,17,19,20,23,26,27,33,39,40,41,46,47,48,52,54,55,59]
		MS	26	[18,20,21,22,24,25,28,29,30,31,32,34,37,38,42,43,44,45,49,50,51,56,57,58]
Main Metabolite Changes	Gut Microbiota Metabolites	Decrease in urinary hippurate	5	[39,41,46,47,48]
		Decrease in urinary *p*-cresol sulphate	2	[41,46]
		Decrease in urinary and faecal SCFAs	10	[30,39,41,47,48,52,54,55,57,58]
		Increase in faecal tyrosine	4	[30,53,55,59]
		Decrease in serological/plasmatic tryptophan	7	[18,24,27,29,35,41,44]
	Compromised Intestinal Barrier	Increase in faecal amino acids	8	[30,52,53,54,55,56,59]
		Decrease in urinary and serological/plasmatic amino acids	19	[16,18,23,24,25,27,29,32,33,34,35,36,39,40,42,45,47,48,49]
	Energy Metabolism Alteration	Decrease in serological/plasmatic, urinary and tissular TCA intermediates	11	[15,18,23,25,34,36,39,40,41,47,48]
		Increase in serological/plasmatic ketone bodies	7	[18,33,34,39,40,41,45]
		Increase in serological/plasmatic glucose	5	[26,34,39,40,41]

## Data Availability

Data sharing not applicable.

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
