# Peer review of "Metabolomics Insights into Inflammatory Bowel Disease: A Comprehensive Review"

_pharmaceuticals, 2021, doi:10.3390/ph14111190_

Round 1
Reviewer 1 Report
Laila et. al. has done a nice piece of review work on IBD “The role of metabolomics unravelling inflammatory bowel disease mechanisms: a comprehensive review”, focusing on metabolomics to explain the underlying mechanism of inflammatory bowel disease (IBD). This is a well written article citing 78 peer-reviewed articles, which is very appropriate and timely.
This reviewer feels that Table 1’s finding should be presented in a graphical manner instead of showing it in main text. The table itself may be included as supplementary table.
Author Response
Thank you very much for your comments and for giving us the opportunity of sending a reviewed version of the manuscript.
We tried to summarize the table in a figure but as the content is quite heterogenous, the figure results quite confusing and with plenty of text. Therefore, we decided to keep it as it was; however, following you suggestion, we included a graphical abstract summarizing the main sections of the document.
We have addressed all the concerns following your suggestions as much as possible. We feel that your comments have considerably improved the final version of the manuscript.

Reviewer 2 Report
Reviewer's report
Title: The role of metabolomics unravelling inflammatory bowel disease mechanisms: a comprehensive review
Version: 1st version
Date: 10.10.2021
Reviewer's report:
The content of the research "The role of metabolomics unravelling inflammatory bowel disease mechanisms" is of considerable interest and relevance. I read the manuscript with enthusiasm; It is well-written, and the results are clearly presented. However, I kindly believe that the method section should be improved and explained in more detail. I recommend authors clearly define all the inclusion and exclusion criteria for the relevant publication. Also, maybe present a diagram of how many publications they had found, how many were eligible, etc.
I believe it is pretty essential to provide precise data on methodology to reach the transparency goal.
If the authors have a registration number for the review, they should also present it.
Author Response
Thank you very much for your comments and for giving us the opportunity of sending a reviewed version of the manuscript. As this is a narrative review, search terms were less restrictive than in a systematic review, and different search terms were included. We defined the main topics related to the search strategy: “Included search terms were related to: (1) metabolomics, (2) IBD, (3) inflammation and/or (4) biomarkers”.
We have modified the inclusion criteria description to a more detailed explanations as well as the exclusion criteria description, according to your suggestions.
We have addressed all the concerns following your suggestions as much as possible. We feel that your comments have considerably improved the final version of the manuscript.

Reviewer 3 Report
The article presented by Laila Aldars-García, Javier P. Gisbert, and María Chaparro, entitled “The role of metabolomics unravelling inflammatory bowel disease mechanisms: a comprehensive review ”, is a review that encapsulates the scientific evidence of metabolomics in elucidating the mechanisms underlying IBD, the changes associated with disease phenotype and therapies, and in identifying new biomarkers of metabolic imbalance in IBD patients. The review is carried out by prestigious authors in IBD.
The article is well written, easy to understand and the vocabulary is correct. The topic is within the scope of the Pharmaceuticals.
First, although it is a bibliographic review and not a systemic one, additional information on the number of articles found in the database, inclusion and exclusion criteria, etc., would be appreciated.
Second, the authors make a table with all the articles found (46 articles), where the characteristics of each of the articles are shown. In my opinion it is too tedious and should go in supplementary tables. The authors should make graphs that group the articles by categories and summarize the findings, e.g., biosamples used with their number of studies, methodology used or most studied metabolites among others. Graphics are more visual, enhance the writing and summarize the findings in a more appropriate way
Thirdly, section 3.2 Methodology to study the metabolome of biological samples, is outside the scope of the work and should be either reduced or eliminated.
Author Response
Thank you very much for your comments and for giving us the opportunity of sending a reviewed version of the manuscript.
- We have included additional information regarding the bibliographic search strategy and the inclusion and exclusion criteria.
- Thank you for your suggestions, we tried to re-structure the table but the criteria for making categories sometimes overlaps certain studies and, in the end, results confusing. Then we decided to keep as it was, however we kindly consider your suggestion if categorizing this type of table for future documents.
- Thank you for your suggestion, we have reduced the length of section 3.2. and made it less specific.
We have addressed all the concerns following your suggestions as much as possible. We feel that your comments have considerably improved the final version of the manuscript.

Reviewer 4 Report
Comments to the Authors of manuscript number: pharmaceuticals-1399074 entitled “The role of metabolomics unravelling inflammatory bowel disease mechanisms: a comprehensive review”.
Inflammatory bowel disease (IBD) is a term for two conditions (Crohn’s disease and ulcerative colitis) that are characterized by chronic inflammation of the gastrointestinal (GI) tract. Prolonged inflammation results in damage to the GI tract. Of course there is between Crohn’s disease and ulcerative colitis. Both are resulted in many social and economic problems as it is mentioned by Authors.
The review-study is very interesting and very well written, and suits to chosen Journal, however, the manuscript should be corrected in some points.
- L 43 – the period for which this cost is calculated should be given.
- L 49-50 What is the meaning of “exposome”?
- L 195-196 Authors should provide short information related to this sentence. In present form is seems to be interrupted thought.
- L 234- how it is associated? Is tryptophan decreased or increased?
- 3.3.2 is there information about the malabsorption of micro- or macro elements? Both amino acids and elements disturbances result in turn in other health problem like osteoporosis (decreased collagen synthesis and mineralization)
- L 282 – the abbreviation should be explained
- L 287 - α-ketoglutarate is a precursor of glutamine, thus it is worth to mention that its decrease leads to the decrease of glutamine, what is mentioned above
- Maybe the better way is to explain all abbreviation in the text, not giving it et the end of text.
Author Response
Thank you very much for your comments and for giving us the opportunity of sending a reviewed version of the manuscript. Please see the response to your suggestions after each point line.
- L 43 – the period for which this cost is calculated should be given.
The period has been included.
- L 49-50 What is the meaning of “exposome”?
It is explained in the previous sentence, “exposure to environmental factors”.
- L 195-196 Authors should provide short information related to this sentence. In present form is seems to be interrupted thought.
Thank you for the suggestion, a previous sentence has been included.
- L 234- how it is associated? Is tryptophan decreased or increased?
A short description of how it is altered, as well as for levoglucosan, was included
- 3.3.2 is there information about the malabsorption of micro- or macro elements? Both amino acids and elements disturbances result in turn in other health problem like osteoporosis (decreased collagen synthesis and mineralization)
Yes, there is information related to gastrointestinal diseases and the consequences of malabsorption. However, we decided not to include this topic because it is broad and the degree of malabsorption depends on how much of the intestine is affected, then it will require a deep and lengthy discussion. In addition, the search strategy did not include this type of studies. However, we appreciate your suggestion and consider for further reviews as it is a very concerning topic in IBD patients.
- L 282 – the abbreviation should be explained
IBD abbreviation was explained in line 33.
- L 287 - α-ketoglutarate is a precursor of glutamine, thus it is worth to mention that its decrease leads to the decrease of glutamine, what is mentioned above
We included α-ketoglutarate in the following sentence “Some TCA metabolites can alter the response of both the innate and adaptive immune systems, including acetyl-CoA, succinate, α-ketoglutarate or fumarate” as it is also involved the immune cells response (glutamine is an essential nutrient for immune cells).
- Maybe the better way is to explain all abbreviation in the text, not giving it et the end of text.
All abbreviations were explained after its first mention in the text, for all the terms included.
We have addressed all the concerns following your suggestions as much as possible. We feel that your comments have considerably improved the final version of the manuscript.

Round 2
Reviewer 2 Report
I thank the authors for their response. I am glad to see that they have considered the revisions, and I believe the manuscript now can be considered for publication.
Author Response
Thank you very much for your revision and response.

Reviewer 3 Report
The authors have not improved the article with the addition of figures/graphs that summarize the work, in my opinion the answer "we tried to restructure the table but the criteria to make the categories sometimes overlap with certain studies and, in the end, it is confusing" is not enough and a review of a Q1 journal requires more effort than summing up the data in a tedious table.
Author Response
Thank you very much for your revision and kind suggestion for the table. We summarized the table in a new table with the main findings. We also included the previous “table 1” as supplementary material.
We have addressed all the concerns following your suggestions as much as possible. We feel that your comments have considerably improved the final version of the manuscript.

Round 3
Reviewer 3 Report
The authors have substantially improved the organization and comprehensiveness of the review.